# An Extended Model for Analyzing the Heat Transfer in the Skin–Microenvironment–Fabric System during Firefighting

**DOI:** 10.3390/ma16020487

**Published:** 2023-01-04

**Authors:** Ying Lei, Faming Wang, Jie Yang

**Affiliations:** College of Safety Science and Engineering, Xi’an University of Science and Technology, Xi’an 710054, China

**Keywords:** heat transfer model, firefighters’ protective clothing (FPC), skin burn, human body movement, fabric periodic movement

## Abstract

This study proposed an extended multi–layer heat transfer model to simulate skin burns of firefighters during firefighting. The proposed model takes into account the effect of fabric movement frequencies, fabric movement amplitudes and human body movement speeds on the heat transfer between the skin and the heat source under low–level radiative exposure. The simulation performance was validated against the simulations in the published literature in terms of the heat transfer in the multi–layer fabric system, skin temperature and skin burns. The results indicated that the fabric periodic movement caused by human body movement decreased the time to skin burns and the skin temperature increased with increasing fabric movement amplitude. During firefighting, the time to 2nd degree burn was 33.3–35.2% shorter at medium human body movement speed than at low and high movement speeds. Furthermore, at low movement speeds, the time to 2nd degree burn was negatively associated with fabric movement amplitude, whereas it was delayed by 12.9–29.8% at the fabric movement amplitude of 2.5 mm at medium and high human body movement speeds. This research provides foundational knowledge for the development of a new generation of firefighters’ protective clothing (FPC) and the assessment of skin burns in firefighters.

## 1. Introduction

Firefighters are typically exposed to radiative environments for extended periods of time while fighting fires, particularly under low–level radiative conditions of 5–20 kW/m^2^ [1]. Thermal energy from the environment is transferred to the skin surface via radiation, conduction and convection, putting firefighters at risk of skin burn injuries, which is a major factor endangering firefighters’ lives. To reduce heat–related illnesses and injuries among firefighters, firefighters’ protective clothing (FPC) must reduce heat transfer from the heat source to the skin. It is especially important to investigate the heat transfer mechanism in the FPC between the skin and its radiative environment, as well as the prediction of skin burns, in order to balance thermal protection and thermal comfort.

In the past decades, bench top (e.g., thermal protective performance and radiant protective performance testers) and full–scale fire manikin tests were widely used to evaluate the thermal protective performance (TPP) of FPC. Lee et al. investigated the effects of heat-exposure intensity and fabric properties (thickness, density and weight) on an FPC’s TPP under nine levels of thermal radiation ranging from 20 to 84 kW/m^2^ [2]. Song et al. used a full–scale fire manikin test under flash fire exposure to determine the skin burn distributions of the human body [3]. Furthermore, the complex effects of the microenvironment between the thermal liner back surface and the skin [4,5], moisture accumulation [6,7,8] and human body state [9] on the TPP were investigated.

Aside from experimental studies, numerical simulation can be used to thoroughly analyze the heat transfer mechanism in an FPC during radiative heat exposure in order to reflect its TPP. Torvi et al. proposed a heat transfer model that predicts the time to skin burns by calculating heat transfer between the fabric, microenvironment and skin under high–level radiative exposure [10]. Su et al. [11] established a multi–layer heat transfer model that takes into account the effects of stored thermal energy discharge in a fabric system on skin burns. Further research considered the effects of the phase change material (PCM) layer [12], anisotropy of materials [13], external compression [14] and heat exposure distance between human body and heat source [15] on heat transfer.

Furthermore, many studies have focused on the microenvironment between the thermal liner back surface and the skin, as well as its effect on heat transfer and skin burns during radiative heat exposure. Ghzay, for example, investigated the positive effect of each microenvironment in a skin–microenvironment–fabric system on thermal protective performance of FPC [16,17,18]. Udayraj calculated convective heat transfer in the microenvironment and investigated the effects of microenvironment thickness, direction (horizontal and vertical) and type (homogeneous and heterogeneous) on skin burns [19]. Deng et al. [20] investigated the effect of a heterogeneous microenvironment with contact and non–contact folds on thermal protective performance of FPC. In general, the studies on heat transfer between the skin and its radiative environment described above focused on a fixed or stationary microenvironment. In fact, human body movement causes a dynamic variation in the thickness of the microenvironment, influencing heat transfer in multi–layer fabric systems and skin burn predictions [21].

There have been few studies that investigate the effects of human body movement on heat transfer in skin–microenvironment–fabric systems. To evaluate the TPP of protective clothing during firefighter movement, Ghazy characterized human body movement using sinusoidal variation of microenvironment thickness [22,23,24]. Udayraj then investigated heat transfer in horizontal and vertical microenvironments during human body movement [25]. Łapka et al. calculated mass transfer in the skin–microenvironment–fabric system during human body movement by accounting for water vapor diffusion transport, adsorption and desorption [26]. The preceding studies either focused on dynamic variation in heat exposure distance or characterized human body movement through fabric periodic movement, but the overall effects remain unknown.

In this study, an extended heat transfer model of the skin–microenvironment–fabric–environment system was proposed, taking into account the comprehensive effects of dynamic variation of the heat exposure distance and fabric periodic movement on heat transfer in the system during firefighter movement. Following that, the proposed model was validated against the published literature in terms of heat transfer and skin burns in a multi–layer fabric system. Finally, the effects of fabric movement frequencies, amplitudes and human body movement speeds on skin burns when firefighters enter and exit the fireground were calculated. This study provides fundamental knowledge for predicting skin burns in firefighters during firefighting.

## 2. Heat Transfer Model of the Skin–Microenvironment–Fabric System

As shown in Figure 1, the skin–microenvironment–fabric system is composed of three components: the FPC (an outer shell, a moisture barrier and a thermal liner), skin tissue (epidermis, dermis and subcutaneous) and the microenvironment between the thermal liner back surface and skin (average thickness of 3 mm).

Regarding heat transfer in the skin–microenvironment–fabric system, the following assumptions were made: in the skin–microenvironment–fabric system, heat transfer is assumed to be one–dimensional and mass transfer is ignored; convective heat transfer only exists on the fabric surface, but thermal radiation can penetrate through a certain depth of the multi–layer fabric system [10]; cooling of the microenvironment during human body movement is ignored [9]; and the thermophysical properties of the outer shell change with temperature.

### 2.1. Heat Transfer between Fabric and Heat Source

Figure 2 depicts a scene of firefighters moving around the fireground while fighting a fire. To calculate radiative heat transfer, the current model considers the heat source and human body to be two parallel coaxial discs with unequal radius. Radiative heat transfer between the two discs is related to their location during radiative heat exposure [27]. Specifically, the incident thermal radiation transferred to the multi–layer fabric system varies dynamically with the heat exposure distance between the human body and the heat source during firefighter movement. The radiation view factors between the heat source, radiative environment and fabric are solved as follows [15]:(1)$$d=\{\;\begin{array}{c} x_{\max}-vt\\ x_{\min}+vt \end{array}\;$$
(2)$$F_{shell\text{-}hs}=\frac{1}{2}\left\{1+\frac{d^{2}{+r}_{hs}{}^{2}}{r_{shell}{}^{2}}\;-\;\sqrt{{\left(1+\frac{d^{2}{+r}_{shell}{}^{2}}{r_{shell}{}^{2}}\right)}^{2}-4{\left(\frac{r_{shell}}{r_{hs}}\right)}^{2}}\right\}$$
(3)$$F_{hs\text{-}shell}=\frac{F_{shell\text{-}hs}A_{fab}}{A_{hs}}$$
(4)$$F_{shell\text{-}amb}=1-F_{shell\text{-}hs}$$
where *d*, *x*_max_ and *x*_min_ are the instantaneous, maximum and minimum distances between the heat source and human body, respectively. v is the human body movement speed. *r_hs_* and *r_shell_* are the heat source and outer shell radius, respectively. *A_hs_* and *A_fab_* are the heat source and outer shell area, respectively.

### 2.2. Heat Transfer in the Multi–Layer Fabric System

The heat transfer equation of the multi–layer fabric system is obtained using the energy conservation law based on the assumptions stated above:(5)$${\left(\rho c_{p}\right)}_{fab}\frac{\partial T}{\partial t}=\frac{\partial }{\partial x}\left(k_{fab}(T)\frac{\partial T}{\partial x}\right)+\frac{\partial q_{rad-absorb}}{\partial x}$$
where *ρ_fab_*, (*c_p_*)*_fab_* and *k_fab_* denote the density, specific heat and thermal conductivity of the fabric, respectively. *q_rad_absorb_* is the absorbed portion of the incident thermal radiation in the multi–layer fabric system, which is calculated using Beer’s law as follows:(6)$$q_{rad-absorb}=q_{rad}(1-\exp(-\gamma_{f}x))$$
(7)$$q_{rad}=\frac{F_{hs-shell}\sigma (\varepsilon_{hs}T_{hs}{}^{4}-\varepsilon_{shell}T_{shell}{}^{4})A_{hs}}{A_{fab}}-\sigma \varepsilon_{shell}F_{shell-amb}(1-\varepsilon_{g})(T_{shell}{}^{4}-T_{amb}{}^{4})$$
where *q_rad_* is the incident thermal radiation transmitted from the heat source to the fabric and *F_hs_shell_* is the proportion of radiation emitted from the heat source intercepted by the fabric. *ε_hs_*, *ε_g_* and *ε_shell_* are the emissivity of the heat source, hot gases and outer shell, respectively. *σ* is the Stephen Boltzmann constant, which is 5.67 × 10^−8^ W/m^2^ K^4^. The extinction coefficient of the outer shell (*γ_f_*) is solved by [10]:(8)γf=−ln (τ)Lshell
where *τ* and *L_shell_* are the transmissivity and the thickness of the outer shell, respectively.

The boundary conditions of the multi–layer fabric system are
(9)$${-k_{shell}(T)\frac{\partial T}{\partial x}|}_{x=0}={q_{rad\_tran}|}_{x=0}-h_{cnv}({T|}_{x=0}-T_{amb})$$
(10)$${-k_{therm}\frac{\partial T}{\partial x}|}_{x=L_{fab}}={q_{therm\_skin}|}_{x=L_{fab}}-k_{air}(T){\frac{\partial T}{\partial x}|}_{x=L_{fab}}$$
(11)$$q_{rad\_tran}=q_{rad}(\exp(-\gamma_{f}x))$$
where, *q_rad_tran_* is the transmitted portion of the incident thermal radiation in the outer shell and *h_cnv_* is the convective heat transfer coefficient between the outer shell and the radiative environment and is calculated using the vertical plate’s natural convection empirical relationship as follows [27]:(12)$$Ra=\frac{g\beta (T_{s}{}_{hell}-T_{a}{}_{mb})L_{f}{}^{3}}{av}$$
(13)$$Nu=\{\begin{array}{l} 0.68+\frac{0.67Ra^{1/4}}{{\left(1+{\left[0.492/Pr\right]}^{9/16}\right)}^{4/9}}\;10^{-1}<Ra<10^{9}\\ 0.825+\frac{0.387Ra^{1/6}}{{\left(1+{\left[0.492/Pr\right]}^{9/16}\right)}^{8/27}}\;Ra>10^{9} \end{array}$$
(14)$$h_{c}{}_{nv}=Nu\frac{k_{a}{}_{ir}}{L_{f}}$$
where *R_a_* (Rayleigh number) is the key parameter to judge natural convection in the microenvironment [28]. *Nu* (Nusselt number) reflects the intensity of convective heat transfer. *L_f_* is the characteristic length of the outer shell, which was 0.1 m.

The microenvironment is assumed to be a rectangular closed cavity, which was used to calculate the radiative heat flux from the back surface of the thermal liner to the skin tissue:(15)$$q_{therm\_skin}=\frac{\sigma (T_{therm}{}^{4}{|}_{x=L_{fab}}-T_{epi}{}^{4}{|}_{x=L_{fab}+L_{air}})}{\left(\frac{1-\varepsilon_{therm}}{\varepsilon_{therm}}+\frac{1}{F_{therm-skin}}\right)+\frac{1-\varepsilon_{skin}}{\varepsilon_{skin}}}$$
where *ε_therm_* and *ε_skin_* are the emissivity of the thermal liner and the skin tissue, respectively. *F_therm-skin_* is the proportion of radiation emitted from the thermal liner back surface intercepted by the skin tissue. The thermal degradation of the outer shell (Nomex) did not occur under the low–level radiative exposure (8.5 kW/m^2^) [29]. Its effective thermal conductivity is weighted by the volume fraction of the fiber and air in the outer shell, as follows:(16)$$k_{fiber}(T)=\{\begin{array}{l} 0.13+0.0018(T-300)\;T<700\;Κ\\ \;1\;T>700\;Κ \end{array}$$
(17)$$k_{air}(T)=\{\begin{array}{l} 0.026+0.000068(T-300)\;T<700\;Κ\\ 0.053+0.000054(T-700)\;T>700\;Κ \end{array}$$
(18)$$k_{shell}(T)=0.8k_{air}(T)+0.2k_{fiber}(T)$$
where *k_fiber_* and *k_air_* are the thermal conductivities of the fiber and the air in the outer shell, respectively.

### 2.3. Heat Transfer in the Microenvironment

The microenvironment was assumed to be a radiation–absorbing medium in the current model. Beer’s law was used to quantify the proportion of thermal radiation absorbed in the microenvironment [18]. The microenvironment heat transfer equation is read as:(19)$${\left(\rho c_{p}\right)}_{air}\frac{\partial T}{\partial t}=\frac{\partial }{\partial x}\left(k_{air}(T)\frac{\partial T}{\partial x}\right)+\frac{\partial q_{rad-absorb1}}{\partial x}$$
where *ρ_air_*, (*c_p_*)*_air_* and *k_air_* are the density, specific heat and thermal conductivity of air, respectively. *q_rad_absorb_*_1_ is the absorbed portion of the radiative heat flux in the microenvironment.

The boundary conditions of the microenvironment are
(20)$${T_{therm}|}_{x=L_{fab}}={T_{air}|}_{x=L_{fab}}$$
(21)$${T_{air}|}_{x=L_{fab}+L_{air}}={T_{epi}|}_{x=L_{fab}+L_{air}}$$

Furthermore, as shown in Figure 3, the sinusoidal variation of microenvironment thickness is used to reflect the effect of fabric periodic movement on heat transfer in the skin–microenvironment–fabric system during human body movement [22].

The instantaneous microenvironment thickness is solved by
(22)$$x=x_{0}+\Delta x\sin(2\pi ft)$$
where *x* and *x*_0_ are the instantaneous and average thicknesses of the microenvironment, respectively; ∆*x* and *f* are the fabric movement amplitude and frequency, respectively.

### 2.4. Heat Transfer in Skin Tissue

Thermal conduction is the primary heat transfer mode in skin tissue [30]. In the current model, the temperature distribution of skin tissue was determined using Pennes’ thermal biological model [31]. The heat transfer equation is written as:(23)$${\left(\rho c_{p}\right)}_{skin}\frac{\partial T}{\partial t}=\frac{\partial }{\partial x}\left(k_{skin}\frac{\partial T}{\partial x}\right)$$

The boundary conditions of the skin tissue are
(24)$${-k_{ep}\frac{\partial T}{\partial x}|}_{x=L_{fab}+L_{air}}=q_{therm\_skin}\exp(-\kappa_{air}L_{air})-k_{air}(T){\frac{\partial T}{\partial x}|}_{x=L_{fab}+L_{air}}$$
(25)$${T_{skin}|}_{x=L_{fab}+L_{air}+L_{skin}}=306.65\;Κ$$
where *ρ_skin_*, (*c_p_*)*_skin_* and *k_skin_* are the density, specific heat and thermal conductivity of skin tissue, respectively. *κ_air_* is the radiation absorption coefficient of the microenvironment, i.e., 5 m^−1^.

The initial temperature of the multi–layer fabric and the microenvironment were both equal to the ambient temperature (300 K). The initial temperature of the skin tissue is linearly distributed from the epidermis to the subcutaneous layer back surface (305.65 K–306.65 K) [32].

### 2.5. Skin Burn Prediction

The skin burn degree is predicted by the most widely used Henriques burn integral [33]:(26)$$\Omega =\int_{0}^{t}P\exp\left(-\frac{\Delta E}{RT(x,t)}\right)\;dt$$
where Ω is the quantitative value of the skin burn degree, *R* is the universal gas constant, which is 8.31 J/(mol·°C), ∆*E* is the activation energy of skin tissue and *P* is the Pre–exponential factor, *t* is the time when the skin burn injury starts (the epidermis–dermis interface temperature reached 44 °C) [30]. The relevant parameters are listed in Table 1.

### 2.6. Numerical Solution

The finite difference method was used to solve the energy equations of the skin–microenvironment–fabric system. As shown in Equation (27), the Crank–Nicholson scheme was used to discretize the energy equations and boundary conditions of each layer of the system. After collation, the algebraic equations with tridiagonal coefficient matrices were obtained. The Thomas method has widely been used to solve algebraic equations with tridiagonal coefficient matrices, which could quickly calculate the temperature at each node via elimination and back substitution [27].

Furthermore, the size of the time and space steps during the numerical solution will affect the results’ convergence and accuracy. As a result, the independence study is shown separately in Figure 4 [28]. The results show that, when the time and space steps are 0.1 s and 5 × 10^−6^ m, respectively, the temperature of the dermis–subcutaneous interface does not change significantly with decreasing step size, so the current step size was chosen. Furthermore, during human body movement, the space step size in the microenvironment varies uniformly with thickness. The temperature obtained at each time step updates the thermophysical properties of the multi–layer fabric system.
(27)$$\left(\rho c_{p})\frac{T_{i}^{n+1}-T_{i}^{n}}{\Delta t}=\frac{1}{2}(k\frac{T_{i+1}^{n}-2T_{i}^{n}+T_{i-1}^{n}}{\Delta x^{2}}+k\frac{T_{i+1}^{n+1}-2T_{i}^{n+1}+T_{i-1}^{n+1}}{\Delta x^{2}})+\gamma_{f}q_{rad}\exp^{}(-\gamma_{f}x\right)$$

## 3. Results and Discussion

### 3.1. Model Validation

To validate the heat transfer model, parameters such as instantaneous heat flux, skin temperature and time to skin burns in single–layer and multi–layer fabric systems with microenvironments under different radiative heat exposures were predicted by the current model and compared with the published literature.

Figure 5 compares the transmitted energy (Figure 5a) in the multi–layer fabric system, stored energy (Figure 5b) and skin temperature (Figure 5c) predicted by the current model and Su et al. [11] under low–level radiative exposure (i.e., 8.5 kW/m^2^). It clearly demonstrates that the current model can accurately simulate heat transfer in a multi–layer fabric system and reflect changes in skin temperature with or without the microenvironment. The time to 2nd degree burn is predicted to be 149.7 s under low–level radiative exposure for 300 s, which is only 0.68% different from the prediction result of Su et al. This could be because the current model does not account for the non–linear variation of the convective heat transfer coefficient between the outer shell and the radiative environment with temperature in order to simplify the numerical solution.

Figure 6 displays the temperature at the epidermis–dermis interface when the multi–layer fabric system with the microenvironment was exposed to a flash fire (83 kW/m^2^) [17]. The results show that the current model overestimates skin temperature during flash fire exposure. The maximum deviation from the skin temperature predicted by Ghazy et al. was less than 1 °C. Table 2 compares the time to skin burns predicted by the current model and Ghazy et al. under stationary (microenvironment thickness 3 mm) and dynamic conditions (frequency: 0.5 s^−1^, amplitude: 1.5 mm) [17,22]. The results showed that the prediction deviation of the time to 2nd degree burn under different conditions was 11.7% and 13.6%, respectively. This could be due to the fact that the microenvironment in the current model is regarded as a radiation participating medium that absorbs thermal radiation. Meanwhile, the periodic cooling of the microenvironment caused by human body movement is ignored. Figure 7 depicts the effect of the single–layer fabric’s thermal shrinkage rate on the epidermis–dermis interface temperature during flash fire exposure [35]. The current model accurately reflects the effect of different thermal shrinkage rates of the single–layer fabric on skin temperature, indicating that during flash fire exposure, skin temperature increases with thermal shrinkage rate. This conclusion, however, was reached without taking into account the effect of thermal shrinkage on the thermal properties of the fabric and the structural integrity of the protective clothing [36]. As a result, the current model can be used to accurately evaluate the thermal protective performance of FPC and skin burn injuries and predict the heat transfer process in single–layer and multi–layer fabric systems under different conditions.

### 3.2. Effects of Fabric Periodic Movement on the Heat Transfer and Skin Burns

During firefighting, firefighters must constantly change their location. The periodic movement of the fabric caused by firefighter movement causes dynamic variation in the thickness of the microenvironment and influences heat transfer in the skin–microenvironment–fabric system. During the simulation, firefighters wearing FPC were exposed to a low–level radiative condition of 8.5 kW/m^2^ for 300 s to further predict the effect of fabric periodic movement on skin burns. The firefighters’ starting point is 0.3 m away from the heat source. During radiative heat exposure, firefighters enter the fireground and move to the location closest to the heat source (0.00005 m) at a low human body movement speed [15].

First, the baseline was the case with no fabric periodic movement during human body movement. When the fabric movement amplitude and frequency were 1.5 mm and 0.25 rps, respectively, the heat transfer of the skin–microenvironment–fabric system was simulated. The results demonstrated that fabric periodic movement caused by human body movement lowers the average temperature of the thermal liner back surface and microenvironment while increasing skin temperature, as shown in Figure 8a,b; this is due to the fabric periodic movement, which causes more thermal energy to be discharged into the skin tissue [24]. Furthermore, the temperature fluctuation caused by the fabric periodic movement vanished at the dermis–subcutaneous interface. When the fabric periodic movement was ignored, the current model predicted a time to 2nd degree burn that was more than 6 s earlier.

Figure 9 shows the effect of fabric periodic movement on the radiative heat flux (Figure 9a), conductive heat flux (Figure 9b) and total heat flux (Figure 9c) through the microenvironment. The results revealed that periodic fluctuations in heat transfer in the microenvironment caused by fabric periodic movement occurred during human body movement. During radiant heat exposure, the average value of the radiative heat flux through the microenvironment decreases. However, the increase in conductive heat flux in the microenvironment outweighs the decrease in radiative heat flux, resulting in greater thermal energy transfer to skin tissue. This microenvironmental heat transfer law explains the variations in the microenvironment and skin temperature. During human body movement, the fabric periodically comes into contact with the skin, causing a rapid increase in conductive heat flux through the microenvironment while decreasing the temperature difference between the thermal liner back surface and the skin. Therefore, the radiative heat flux in the microenvironment decreased as the temperature difference decreased.

Second, the amplitude of fabric movement during human body movement varies with the size and distribution of the microenvironment between the thermal liner back surface and skin at local body locations [37,38]. As a result, skin burns at various fabric movement amplitudes during human body movement should be quantified. In the current model, the fabric movement amplitudes were varied from 0 to 2.5 mm.

Figure 10 depicts the effect of fabric movement amplitude on heat flux through the microenvironment at low human body movement speed. The results show that the amplitude of the fabric movement increased the fluctuation of the radiative (Figure 10a) and conductive (Figure 10b) heat flux in the microenvironment. Despite the fact that the larger fabric movement amplitude resulted in lower radiative heat flux in the microenvironment, the total heat flux (Figure 10c) delivered to the skin increased. This is because the multi–layer fabric system approaches closer to the skin as the fabric movement amplitude increases during human body movement, resulting in greatly reduced conductive resistance in the microenvironment. As a result, thermal conduction transferred more thermal energy to the skin surface. Variations in fabric movement amplitude are more important for conductive heat transfer in the microenvironment [24]. Furthermore, during the initial stage of heat exposure, the heat flux transferred to the skin tissue through the microenvironment was less than 0 kW/m^2^. This could be due to the thermal energy discharge of the human body caused by the initial temperature of the skin being higher than the back surface temperature of the thermal liner.

The effect of fabric movement amplitude on temperature distribution in the skin–microenvironment–fabric system is illustrated in Figure 11. The temperature of the thermal liner back surface and microenvironment decreased as the fabric movement amplitude increased (Figure 11a,b), whereas the skin temperature increased (Figure 11c,d). This is consistent with the microenvironmental heat transfer law. Temperature fluctuations at the thermal liner back surface, microenvironment and epidermis–dermis interface increased as the amplitude of fabric movement during human body movement increased. Furthermore, a fabric movement amplitude of less than 1 mm causes periodic fluctuations in heat transfer in the microenvironment but has no effect on skin temperature.

### 3.3. Effects of Human Body Movement Speed on the Heat Transfer and Skin Burns

To study heat transfer in the skin–microenvironment–fabric system at different movement speeds, the current model divides the human body movement speed into three levels: low (300 s), medium (100 s) and high (60 s), based on the time it takes the firefighter to reach the location closest to the heat source [15]. At different movement speeds, the corresponding fabric movement frequencies (0.25, 0.5 and 1rps) are assumed [26]. As the human body movement speed increases during radiative heat exposure, the firefighter returns from the fireground and remains at the initial point. The firefighter state at high movement speed is plotted in the following figures.

Figure 12 presents the trends in total heat flux through the microenvironment at various human body movement speeds with and without fabric periodic movement. As the firefighters moved slowly into the fireground, the total heat flux through the microenvironment increased. Furthermore, the total heat flux through the microenvironment fluctuated at first but gradually stabilized as the speed of human body movement increased. This could be due to dynamic changes in the heat exposure distance between the human body and the heat source. As the firefighter exits the fireground, the increasing rate of incident thermal energy transferred to the fabric system gradually decreases and eventually stabilizes at a low value [15]. As a result of the low temperature difference between the thermal liner back surface and skin, the total heat flux in the microenvironment is stabilized. Furthermore, due to the delay in heat transfer, the total heat flux through the microenvironment increased rapidly during the initial stage of the firefighter exiting the fireground.

Figure 13 illustrates the temperature distribution in the skin–microenvironment–fabric system at various human body movement speeds with and without fabric periodic movement. The temperature variation at the thermal liner back surface and microenvironment, as shown in Figure 13a,b, follows the same pattern as the total heat flux in the microenvironment. When firefighters enter the fireground, the temperature of the thermal liner back surface, microenvironment and skin rises rapidly and the rate of rise rises with the speed of human body movement. The skin temperature increased during the radiative heat exposure, as shown in Figure 13c,d; a decrease in the rate of temperature increase is clearly visible when the firefighters leave the fireground. Skin burn injuries occur more quickly when the human body moves quickly. Furthermore, as the frequency of fabric movement increased, the temperature at the epidermis–dermis interface and the dermis–subcutaneous interface increased, contradicting the positive effect of fabric movement frequency reported in the literature [24]. This is because the current model considers the case of no ambient cold air during radiative heat exposure[9]; thus, the frequent cooling phenomenon of the microenvironment at a higher fabric movement frequency is ignored.

Figure 14 depicts the skin burn predictions at various human body movement speeds while taking into account fabric periodic movement. The results revealed that the time to skin burns initially decreased but gradually increased as human body movement speed increased during firefighting. When compared to low and high movement speeds, the time to 1st and 2nd degree burns was reduced by 28.4–33.8% and 33.3–35.2%, respectively. Su et al. [15] found a similar variation trend in the time to skin burns at different movement speeds. Although skin burn injuries occurred more quickly at high movement speeds, the accumulation of incident thermal energy was reduced because the firefighter stayed away from the heat source for an extended period of time during radiative heat exposure. At a high movement speed, this may result in a larger time difference between 1st and 2nd degree burns.

Figure 15 predicts the effect of fabric movement amplitude on skin burns at various human body movement speeds. The results show that the time to skin burns decreased as the fabric movement amplitude increased at low movement speed and the decreasing rate increased. At medium and high movement speeds, the time to skin burns was delayed at a fabric movement amplitude of 2.5 mm. When compared to a fabric movement amplitude of 2 mm, the time to 1st and 2nd degree burns was delayed by 14.3–20.3% and 12.9–29.8%, respectively. Furthermore, the current model predicts the effect of fabric movement amplitude on skin burns when the dynamic variation of heat exposure distance between the human body and the heat source is ignored. The results revealed that the time to 2nd degree burn decreased as the amplitude of fabric movement during human body movement increased. As a result, the fluctuation of the time to skin burns in Figure 15b,c at a fabric movement amplitude of 2 mm is related to the dynamic variation of heat exposure distance between the human body and the heat source.

## 4. Limitations

This study simulated heat transfer, temperature distribution and skin burns in a skin–microenvironment–fabric system at various human body movement speeds during firefighting; however, there are some limitations. First, the proposed conjugated conductive–radiative heat transfer model did not account for forced convective heat transfer in the microenvironment during human body movement, which may result in some differences in skin burn predictions. Second, sweat and moisture in the environment have complex effects on heat transfer in a multi–layer fabric system. The proposed heat transfer model will incorporate mass transfer in the multi–layer fabric system. Finally, the proposed model assumed a relationship between human body movement speed and fabric movement frequency, which will be improved and validated in real–time scenarios in future studies.

## 5. Conclusions

Heat transfer and skin burn at local body locations during firefighting were numerically investigated in this study using the proposed heat transfer model. The dynamic variation in the heat exposure distance and the fabric movement amplitude and frequency at different human body movement speeds were specifically incorporated into the proposed model. The following are the conclusions:(1)Fabric periodic movement reduced the time to skin burns during human body movement compared to no fabric movement and skin temperature increased as fabric movement amplitude increased.(2)When the firefighters entered the fireground, the rate of temperature increase in each layer of the skin–microenvironment–fabric system increased with the speed of human body movement. The skin burn injury began with rapid movement.(3)Human body movement speed along the same movement path has a significant effect on skin burn. The current model reduces the time to 2nd degree burn by 33.3–35.2% under medium movement speed.(4)At low movement speeds, the time to skin burns was negatively correlated with fabric movement amplitude. With an increase in fabric movement amplitude, the time to 2nd degree burn was reduced by 7.9%. When the fabric movement amplitude was 2.5 mm at medium and high movement speeds, the negative effect of fabric movement amplitude on skin burn was eliminated. Reduced time to 2nd degree burn was by 12.9–29.8% when compared to 2 mm fabric movement amplitude.

## Figures and Tables

**Figure 1 materials-16-00487-f001:**
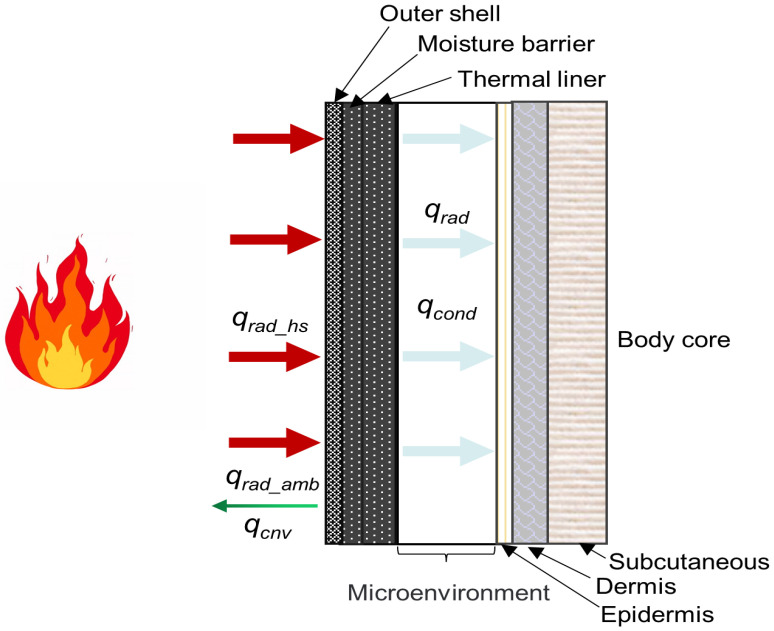
Schematic diagram of heat transfer mechanism in the skin–microenvironment–fabric system.

**Figure 2 materials-16-00487-f002:**
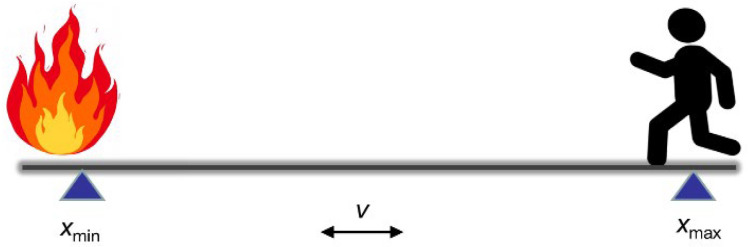
Schematic diagram of the firefighting scene.

**Figure 3 materials-16-00487-f003:**
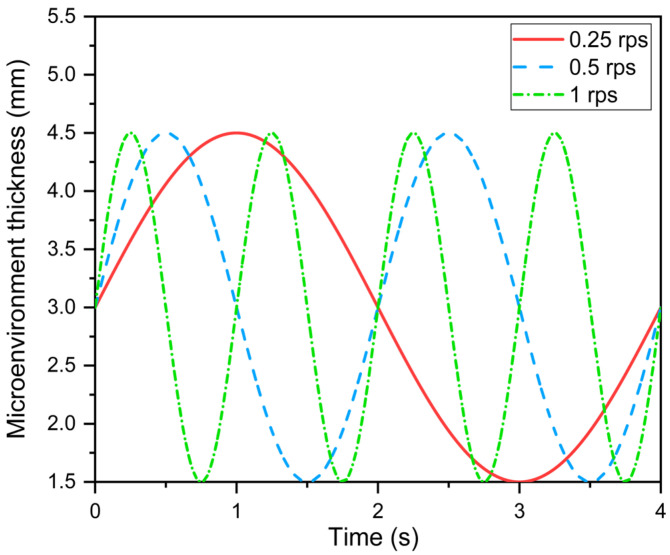
Sinusoidal variation of the microenvironment thickness.

**Figure 4 materials-16-00487-f004:**
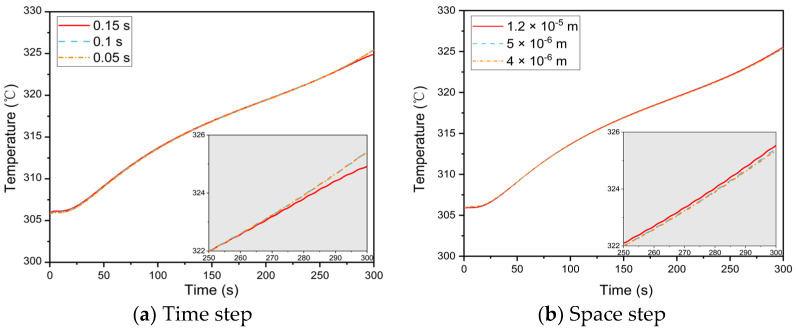
Simulation results of the grid independence study.

**Figure 5 materials-16-00487-f005:**
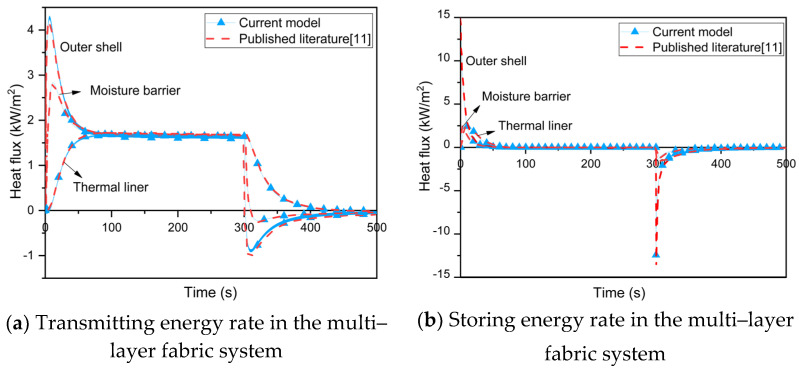
Comparison between the current model and the published literature [11].

**Figure 6 materials-16-00487-f006:**
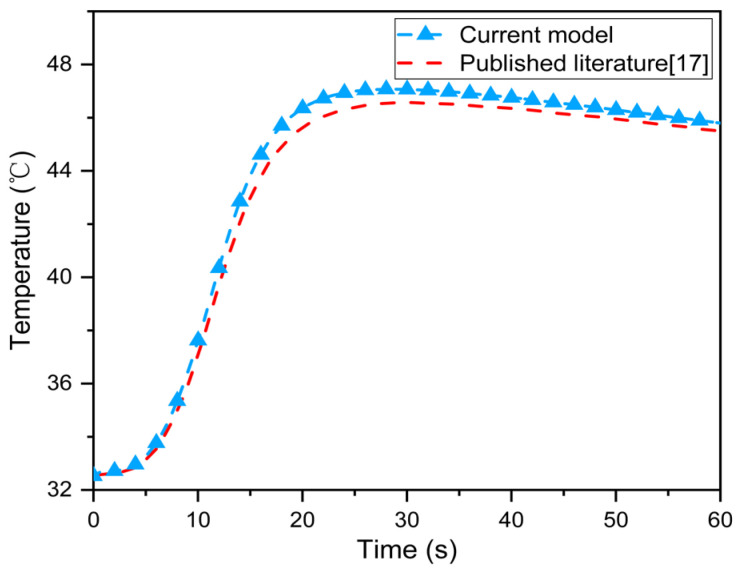
Comparison between the temperature at epidermis–dermis interface predicted by the current model and the published literature [17].

**Figure 7 materials-16-00487-f007:**
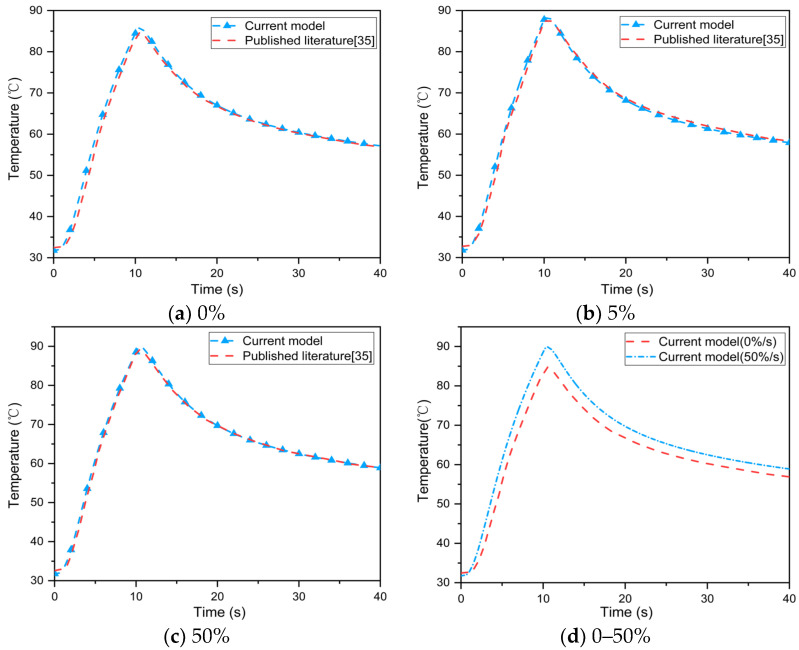
Comparison between the temperature at epidermis–dermis interface under different thermal shrinkage rates predicted by the current model and the published literature [35].

**Figure 8 materials-16-00487-f008:**
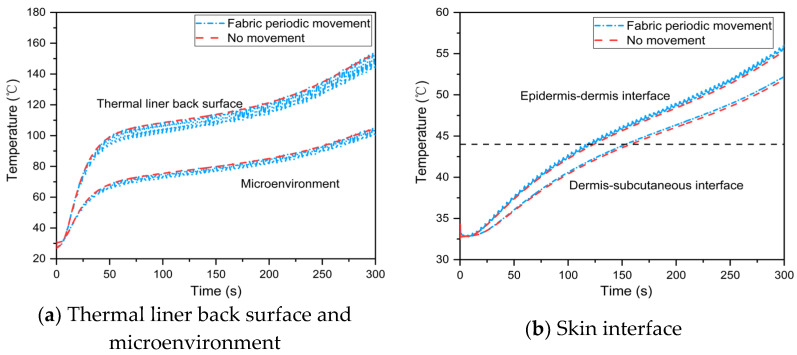
Effect of the fabric periodic movement on temperature distribution in the skin–microenvironment–fabric system.

**Figure 9 materials-16-00487-f009:**
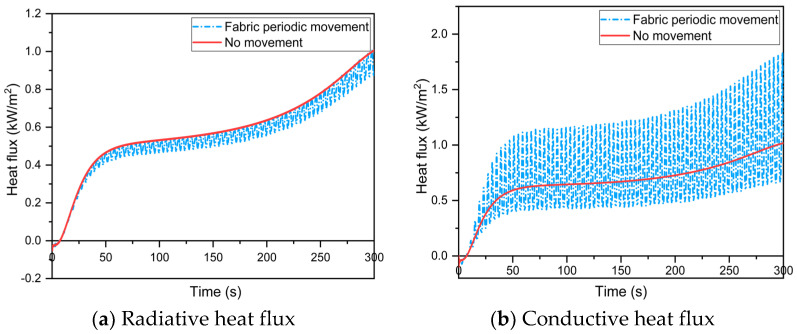
Effect of the fabric periodic movement on heat flux through the microenvironment between the thermal liner back surface and the skin.

**Figure 10 materials-16-00487-f010:**
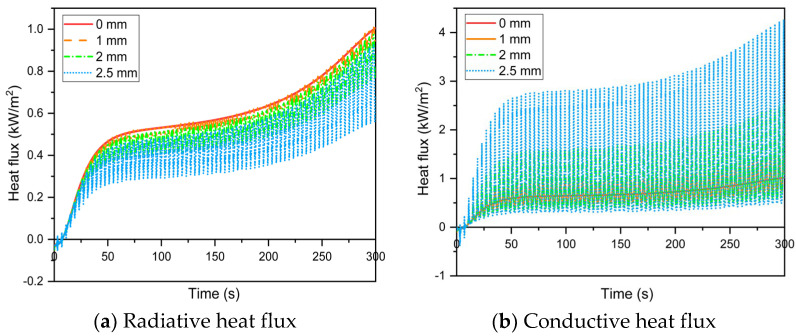
Effect of the fabric movement amplitude on radiative, conductive and total heat fluxes through the microenvironment between the thermal liner back surface and the skin.

**Figure 11 materials-16-00487-f011:**
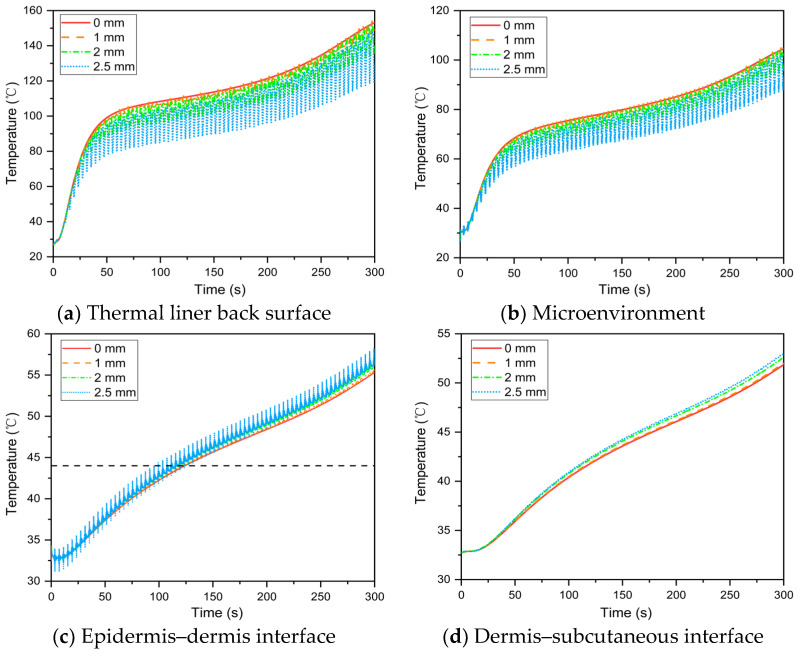
Effect of fabric movement amplitude on the temperature distribution in the skin–microenvironment–fabric system.

**Figure 12 materials-16-00487-f012:**
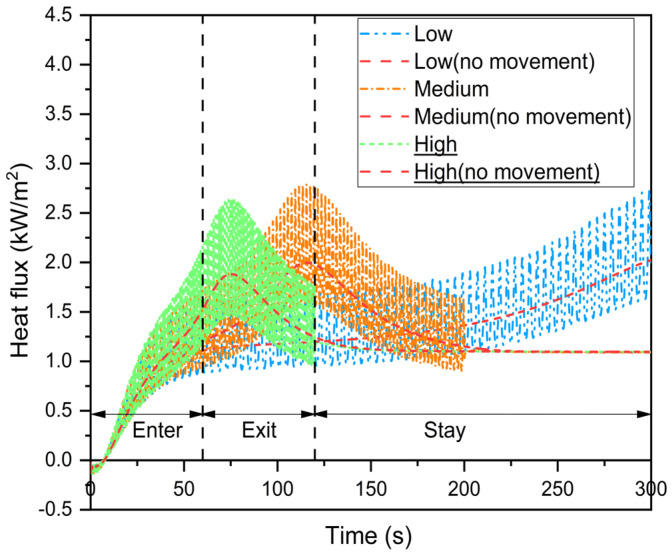
Effect of the human body movement speed on total heat flux through the microenvironment between the thermal liner back surface and the skin.

**Figure 13 materials-16-00487-f013:**
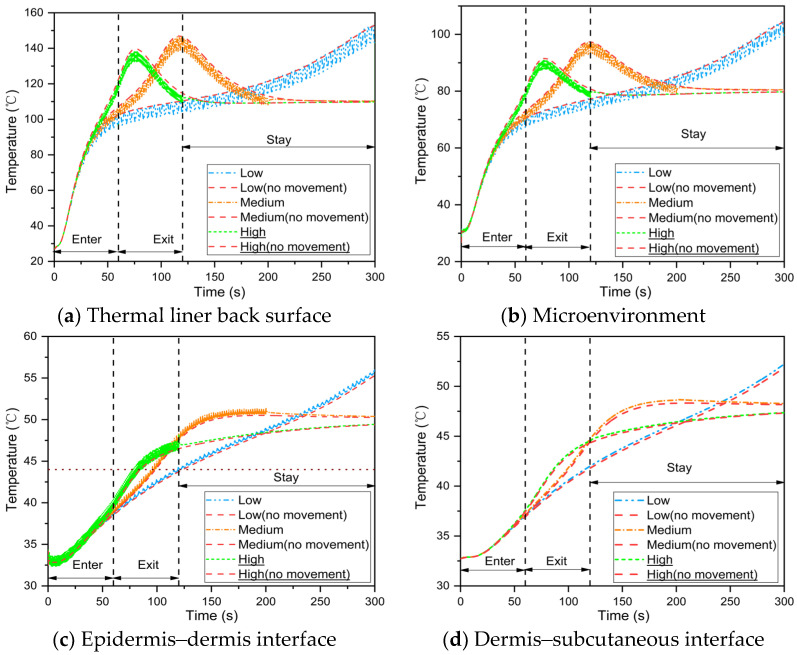
Effect of human body movement speed on the temperature distribution in the skin–microenvironment–fabric system.

**Figure 14 materials-16-00487-f014:**
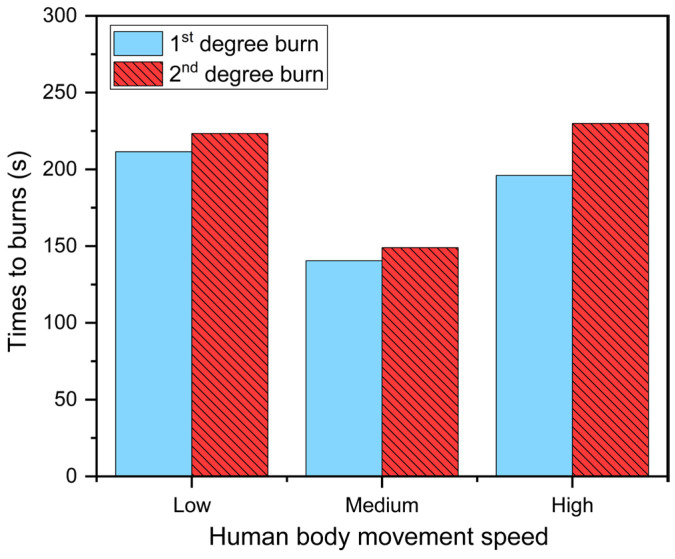
Effect of the human body movement speed on skin burns.

**Figure 15 materials-16-00487-f015:**
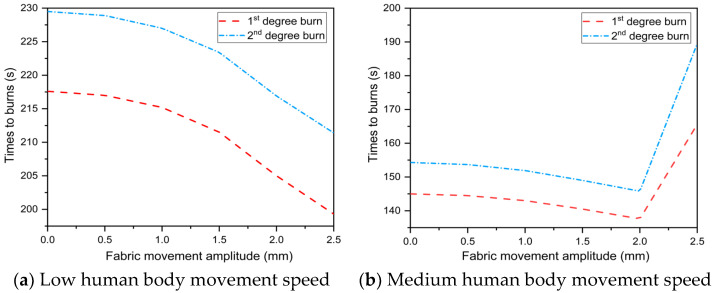
Effect of the fabric movement amplitude on skin burns at different levels of human body movement speed.

**Table 1 materials-16-00487-t001:** Values of *P* and ∆*E/R* in the Henriques burn integral [34].

Skin	Temperature (°C)	P (s^−1^)	∆*E/R* (K)
Epidermis–dermis interface	44 ≤ *T* ≤ 50	2.185 × 10^124^	93,534.9
	*T* > 50	1.823 × 10^51^	39,109.8
Dermis–subcutaneous interface	44 ≤ *T* ≤ 50	4.32 × 10^64^	50,000
	*T* > 50	9.39 × 10^104^	80,000

**Table 2 materials-16-00487-t002:** Time to 2nd degree and 3rd degree skin burn injuries in both stationary and dynamic microenvironments predicted the current model and models reported in published literature [17,22].

	Time to 2nd Degree Burn (s)		Time to 3rd Degree Burn (s)	
Microenvironment	Stationary	Dynamic	Stationary	Dynamic
Published literatures [17,22]	5.3	5.3	26.6	25.7
Current model	4.68	4.58	22.1	21.34
Relative error	11.7%	13.6%	16.9%	17%

## Data Availability

Not applicable.

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
