# Peer review of "An Extended Model for Analyzing the Heat Transfer in the Skin–Microenvironment–Fabric System during Firefighting"

_materials, 2023, doi:10.3390/ma16020487_

Round 1

Reviewer 1 Report

The present reviewer believes that the submitted ms can be published after some corrections:  

1.     What is the novelty of the present work? Please mentioned at the end of introduction section clearly.

2.     How the grid size affect the results? Please add grid independency section.

3.     Why the time step is 0.1 s? Does the time step affect the results? Please show the results for the smaller and bigger time step.

4.     In the validation section, it is recommended to add a table for showing the error in percentage form.

5.     It is recommended to add some papers about heat transfer in the introduction section:

10.4208/cicp.OA-2020-0001

10.1007/s11630-021-1517-1

Reviewer 2 Report

Dear Aythors,

The manuscript entitled An extended model for analyzing the heat transfer in the skin-microenvironment-fabric system during firefighting represents a valuable study to the field. I think the authors have descibed successfully all the aspects of the current study, representing a valuable work.

I do not have anything else to add as comments for the current manuscript. This study is very interesting and i want to congratulate the authors for their work. Well done!!

Author Response

Dear  reviewer:

    Thank you very much for your careful reviews.

    Thank you very much for your support for the current study, and hope that we will have more academic communication.

    Thank you very much.

     Sincerely Yours,

     J.Yang

     Dec. 15, 2022

    Authors: Ying Lei, Faming Wang, Jie Yang

    Title: An extended model for analyzing the heat transfer in the skin-            microenvironment-fabric system during firefighting

    Manuscript ID: materials-2056729

Reviewer 3 Report

This is an interesting work but it requires some further improvements as listed below:

The abstract is well drafted

Line 141- 142 requires a citation

Line 167 requires a citation after radiative exposure value !

Line 204 required details of value for each value not only the sum

Not clear how does work this method “Thomas method.” Please give some brief background in order to replicate it

Overall the numerical solution is the weakest part of this work as now

It seems your model converge very well with the literature (Fig 4) my question what is different in your work in order to consider this novel !

I strongly advice to use better font size for the graphs axes, as for example Figure 8 are very small and difficult to read other graphs in other figures were noted the same

The conclusion part can be revised to indicate more quantitative details and improvements as now is very qualitative

Round 2

Reviewer 3 Report

.